# Leveraging molecular structure and bioactivity with chemical language models for de novo drug design

Michael Moret [1], Irene Pachon Angona [1], Leandro Cotos[1], Shen Yan [2], Kenneth Atz [1], Cyrill Brunner[1], Martin Baumgartner [2], Francesca Grisoni [1,3,4] ✉ & Gisbert Schneider [1,5] ✉

Generative chemical language models (CLMs) can be used for de novo molecular structure generation by learning from a textual representation of molecules. Here, we show that hybrid CLMs can additionally leverage the bioactivity information available for the training compounds. To computationally design ligands of phosphoinositide 3-kinase gamma (PI3Kγ), a collection of virtual molecules was created with a generative CLM. This virtual compound library was refined using a CLM-based classifier for bioactivity prediction. This second hybrid CLM was pretrained with patented molecular structures and fine-tuned with known PI3Kγ ligands. Several of the computer-generated molecular designs were commercially available, enabling fast prescreening and preliminary experimental validation. A new PI3Kγ ligand with sub-micromolar activity was identified, highlighting the method's scaffold-hopping potential. Chemical synthesis and biochemical testing of two of the top-ranked de novo designed molecules and their derivatives corroborated the model's ability to generate PI3Kγ ligands with medium to low nanomolar activity for hit-to-lead expansion. The most potent compounds led to pronounced inhibition of PI3K-dependent Akt phosphorylation in a medulloblastoma cell model, demonstrating efficacy of PI3Kγ ligands in PI3K/Akt pathway repression in human tumor cells. The results positively advocate hybrid CLMs for virtual compound screening and activity-focused molecular design.

Computational methods have become key players in hit and lead discovery in pharmaceutical research, complementing experimental high-throughput screening[1]. Bespoke virtual compound libraries provide access to untapped regions of the chemical space[2], thereby extending the diversity of potential drug candidates. However, owing to the potentially unlimited size of virtual chemical libraries, concerns have been raised over the pragmatism of successfully screening billions of molecules virtually with a potentially high risk of false positives[2,3]. To mitigate some of these challenges, researchers have employed generative deep learning models to construct compounds on demand by de novo design and to obtain small, focused virtual compound libraries[4,5]. A variety of data-driven approaches can be used to generate focused virtual chemical libraries and create molecules with the desired properties[5–18]. Chemical language models (CLMs) are

[1]ETH Zurich, Department of Chemistry and Applied Biosciences, Vladimir-Prelog-Weg 4, 8093 Zurich, Switzerland. [2]University of Zurich, University Children's Hospital, Children's Research Center, Pediatric Molecular Neuro-Oncology Research, Lengghalde 5, 8008 Zurich, Switzerland. [3]Eindhoven University of Technology, Institute for Complex Molecular Systems and Eindhoven Artificial Intelligence Systems Institute, Department of Biomedical Engineering, Groene Loper 7, 5612AZ Eindhoven, The Netherlands. [4]Center for 393 Living Technologies, Alliance TU/e, WUR, UU, UMC 394 Utrecht, Utrecht 3584 CB, The Netherlands. [5]ETH Singapore SEC Ltd, 1 CREATE Way, #06-01 CREATE Tower, Singapore 138602, Singapore. ✉e-mail: f.grisoni@tue.nl; gisbert@ethz.ch

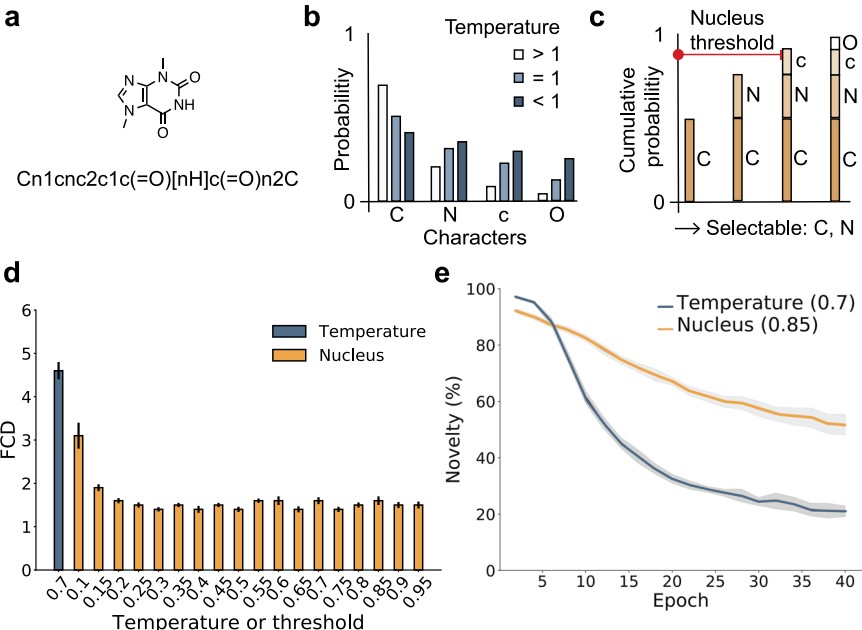

**Fig. 1 | De novo molecular generation with the CLM. a** SMILES string representation of a molecule. **b** Example of the effect of the temperature parameter on the probability distribution learnt by the CLM. **c** Example of the effect of the nucleus sampling threshold. Only the characters N and C can be sampled here. **d** Fréchet ChemNet Distance (FCD) comparison between temperature and nucleus sampling after the pretraining (reported as the mean with standard deviation over 10 repeats with 5000 molecules sampled per repeat). **e** Comparison of the novelty of the generated SMILES strings during the transfer learning between temperature sampling (temperature = 0.7) and nucleus sampling (threshold = 0.85). Mean values (lines) and standard deviations (shaded areas) are shown for ten repeats (1000 SMILES strings were sampled every second epoch over 40 epochs). Novelty is expressed as the percentage of SMILES strings generated that were valid and not included in either the training or the fine-tuning data.

based on neural networks for processing string representations of molecules (e.g., simplified molecular input line entry system [SMILES] strings; Fig. 1a)[5,7,19]. CLMs have already been successfully employed to generate focused virtual chemical libraries. Examples of de novo-designed bioactive molecules include inhibitors of vascular endothelial growth factor receptor 2 (VEGFR-2) pathway[7], as well as nuclear hormone receptor modulators[20–23].

The creation of a focused virtual chemical library with a CLM generally includes three basic steps: (i) model pretraining with a large set of molecules to learn the SMILES grammar and the feature distribution of the pretraining data, (ii) transfer learning with a smaller set of molecules (fine-tuning set) to bias the molecule generation by the CLM toward the chemical space of interest, and (iii) sampling of new molecules from the data distributions modeled in steps (i) and (ii)[5,24]. There are alternative approaches for CLM development, e.g., model fine-tuning (step ii) by reinforcement learning[6,25].

In this study, we developed a data-driven molecular design pipeline that leverages both the structural and bioactivity information of known ligands to generate bespoke molecules by learning from a textual representation of molecules. We pretrained two CLMs, each with a distinct pretraining strategy, on a large set of patented compound structures (one for molecular generation and one for classification). Both CLMs were fine-tuned on inhibitors of phosphoinositide 3-kinase gamma (PI3Kγ), which is an anticancer, anti-inflammatory, and immunomodulatory drug target[26,27]. For rapid validation, commercially available compounds from the set of de novo-generated molecules were tested first, as opposed to synthesizing them, revealing a new ligand of phosphoinositide 3-kinase gamma (PI3Kγ) with sub-micromolar activity. This result confirmed the scaffold-hopping capability of the de novo molecular design pipeline. In addition, two of the top-ranked de novo-generated molecular designs and several derivatives were synthesized. These compounds potently inhibited PI3Kγ activity, corroborating the applicability of the computational approach to hit-to-lead optimization.

## Results and discussion

Molecular design and scoring were performed in two steps, each of which was executed by a distinct CLM: (i) molecular de novo design and (ii) refinement of the generated virtual molecule library using the available ligand bioactivity data for the target of interest (PI3Kγ).

### Focused library generation

**Chemical language model.** A CLM based on a long short-term memory (LSTM) neural network and SMILES strings as input was developed for the de novo generation of a focused virtual chemical library for PI3Kγ[28]. To learn from unlabeled data, CLMs leverage "self-supervised" learning[29]. Specifically, the CLM was trained with an autoregressive approach, i.e., the process of iteratively predicting the next character in a SMILES string given all the previous characters in the string (Fig. 2a)[30]. In previous studies, CLMs were pretrained on molecules with known biological activity ($IC_{50}$, $EC_{50}$, $K_d$, and $K_i$) <1 μM retrieved from the ChEMBL database[20,23,31–33]. Although the training set can capture the general features of bioactive compounds, it does not necessarily represent the physicochemical properties of approved drugs. Here, to enable the CLM to capture features related to approved drugs, we used 839,674 molecules from the US patent database for pretraining[34]. We hypothesized that patented compounds are more likely to become marketed drugs than the bioactive molecules deposited in ChEMBL. Transfer learning was performed to properly focus the pretrained CLM toward the target space of PI3Kγ ligands. For transfer learning, 46 PI3Kγ inhibitors with $IC_{50} \leq 100$ nM were selected from the Drug Target Commons (DTC) database[35].

**Nucleus sampling for molecule generation.** CLMs generate new molecules by extending strings from a "start" character until the "stop" character is sampled or when reaching a preset maximum string length. String characters are iteratively added by weighted random sampling from the probability distribution learned by the CLM during training. The more likely a given character is at a given step according to the

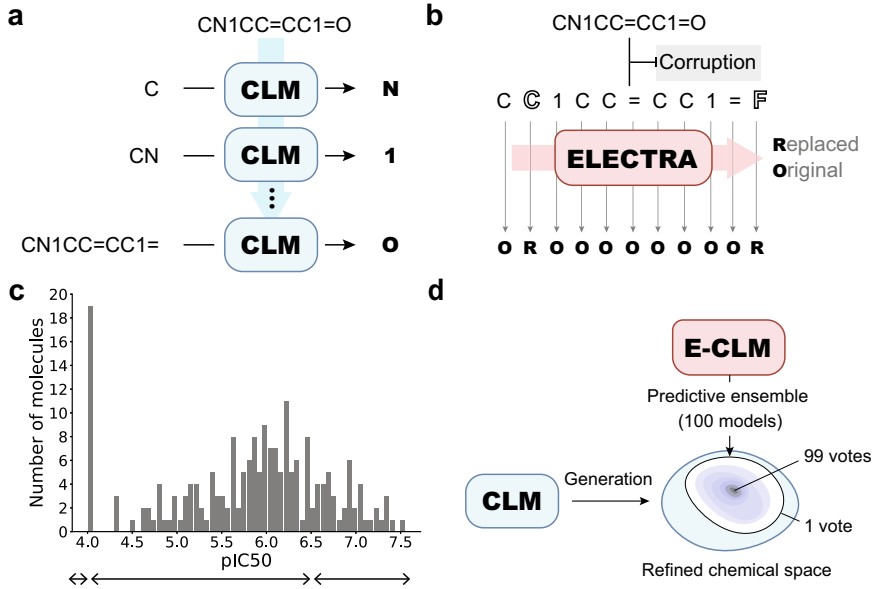

**Fig. 2 | Bioactivity prediction. a** A CLM for molecule generation iteratively predicts the next character in a SMILES string given the preceding characters ("autoregressive" approach). **b** An E-CLM (a CLM pretrained with the ELECTRA method) is trained on corrupted SMILES strings aiming to predict, for each string character, whether it is the original (correct) or a corrupted (substituted) character. **c** Activity distribution of the PI3Kγ ligands. Compounds with annotated $pIC_{50} \leq 4.0$ were considered "inactive", and a $pIC_{50}$ value of 6.5 was used to separate the "moderately active" from the "highly active" compounds. **d** The molecular structures (in the form of a SMILES string) of the fine-tuning set were used to focus the CLM (pretrained on the US patent database) on the chemical space of the target of interest (PI3Kγ). To account for the uncertainty in the predictions, we employed an ensemble of 100 models to rank the generated molecules by the number of "votes".

probabilities learned by the CLM, the more often it will be sampled, and vice versa. Narrowing the probabilities learned by the CLM with a parameter (the so-called temperature; Fig. 1b) generally improves the sampling process[31]. This improvement occurs in terms of (i) the quality of the SMILES strings generated, as reflected by their validity (grammatically valid SMILES strings), uniqueness (nonrepetitive molecules), and novelty (molecules not present in the pretraining and fine-tuning data) and (ii) the similarity of the sampled virtual chemical libraries to the reference data in terms of their chemical structures and predicted bioactivities, as measured by the Fréchet ChemNet Distance (FCD)[36]. However, with this "temperature sampling" approach, unlikely SMILES characters can be sampled, which could result in the construction of molecules that do not match the design objectives. Here, aiming to prevent the CLM from picking unlikely SMILES characters by temperature sampling, we employed "nucleus sampling"[37]. This method reflects the confidence of the model in its predictions by allowing only the most probable character(s) to be sampled using a probability threshold based on the cumulative probabilities of the SMILES characters (Fig. 1c).

Nucleus sampling improved upon temperature sampling in terms of lower FCD values (Fig. 1d), indicating a greater overall similarity of the de novo-generated molecules to the pretraining set in terms of structural and bioactivity properties. During transfer learning, nucleus sampling generally improved the quality of the sampled molecules in terms of the novelty of the SMILES strings compared to the best temperature sampling data obtained (Fig. 1e)[33]. The results were stable over a range of sampling threshold values (Supplementary Table 1). However, nucleus sampling did not outperform temperature sampling in terms of the uniqueness, validity, and novelty of the SMILES strings generated after the pretraining (Supplementary Table 2). To create a PI3Kγ focused chemical library during transfer learning, we used nucleus sampling with a threshold of 0.85. A total of 5000 SMILES strings were sampled over 50 transfer learning epochs with ten repetitions (5000 × 50 × 10). A total of 2,500,000 SMILES strings were generated, of which 1,121,735 were valid, unique, and novel (nonidentical) compared to both the training and fine-tuning compounds.

## Bioactivity prediction with a hybrid chemical language model
**Leveraging bioactivity data for molecule selection.** The availability of bioactivity data for the fine-tuning molecules permitted the training of a bioactivity prediction model to select the most promising de novo designs[38]. Chemoinformatics methods often rely on precomputed features (molecular descriptors), combined with a machine learning algorithm for molecular property prediction. In this study, we aimed to explore the potential of a SMILES string-based hybrid CLM to predict bioactivity. This neural network model combines a generative CLM with a classifier network. Given that (i) inactive molecules on PI3Kγ were annotated with $pIC_{50} = 4.0$ (Fig. 2c) and (ii) there is a natural ordering of the PI3Kγ ligands according to their $pIC_{50}$ values, the bioactivity prediction task was framed as an ordinal classification task, i.e., classification with a class order[39]. Such a model considers both the active and inactive compounds for training and preserves both the class labels and the class order. For model training, we defined three class labels: "inactive" ($pIC_{50} \leq 4.0$, 34 molecules), "moderately active" ($4.0 < pIC_{50} \leq 6.5$, 121 molecules), and "highly active" ($pIC_{50} > 6.5$, 43 molecules). The CLM generated a focused virtual chemical library by leveraging the structural information of the molecules used for fine-tuning, while the classifier layer factored their activity labels into the model (Fig. 2d).

We explored two different pretraining strategies for feature learning with a large amount of unlabeled data.

1. Autoregressive pretraining (Fig. 2a). This strategy is analogous to the one performed for the generative CLM.
2. ELECTRA (efficiently learning an encoder that classifies token replacements accurately) pretraining (Fig. 2b)[40]. The ELECTRA approach is based on training a model to distinguish between "real" input characters and "corrupt" ones, which was previously shown to be useful for contextual representation of natural language[40]. We adapted ELECTRA for the CLM training with an LSTM model and SMILES strings as input[28]. The training data contained corrupted input SMILES strings generated by randomly substituting multiple characters with other characters of the

SMILES language. The CLM was trained to spot these corrupted characters.

We hypothesized that, compared to autoregressive pretraining, ELECTRA pretraining has a more appropriate inductive bias (i.e., the set of algorithmic assumptions to solve a given task) to extract useful features for ordinal classification. The inductive bias of autoregressive pretraining is particularly suited for generating SMILES strings because the training and generative tasks are the same, namely, adding characters iteratively. However, ligands of the same macromolecular target tend to have similar chemical substructures, and, therefore, the ability of a model to distinguish small structural changes was deemed relevant. At the same time, small structural changes might lead to a drastic variation in biological activity (the so-called activity cliffs[41]). Hereinafter, the model that was pretrained with the ELECTRA method is referred to as "E-CLM".

To probe the effect of the pretraining scheme on the predictions, we added only a single feedforward layer to the pretrained CLM and E-CLM for bioactivity prediction. This additional network layer consisted of three neurons, one for each of the three bioactivity classes. It was added to fine-tune the entire network for bioactivity prediction[42,43]. To mitigate the class data imbalance, we applied oversampling to the classes with fewer data (i.e., the "inactive" and "highly active" classes)[44].

Overall, we found that the E-CLM performed better than the standard CLM for the task of identifying the most active molecules, while minimizing the number of inactive molecules misclassified as "highly active". For the chosen threshold (0.4), the E-CLM had a false positive rate of 10.0% compared to 46.7% for the CLM for the same true positive rate (71.3%) (Supplementary Figs. 2a, 3a). Fine-tuning of all neural network weights performed better than keeping the weights of one of the two layers constant (Supplementary Fig. 2a, c, d). These results highlight the importance of choosing an appropriate pretraining method for cheminformatic applications, depending on the downstream task, e.g., data generation or classification.

**Increasing the prediction confidence using deep ensemble learning.** Deep learning models suffer from a decrease in performance when applied to out-of-domain data[45], a well-known issue in quantitative structure-activity relationship modeling[46,47]. To increase the confidence in the bioactivity predictions, we used a deep ensemble model by combining the predictions of multiple models with a majority voting approach[48,49]. Owing to the nondeterministic optimization process, repeats of the same CLM training procedure will lead to different models. Deep ensemble learning has been shown to perform well across different domains to account for the predictive uncertainty of the models, while having the benefit of being straightforward to implement[50]. Accordingly, 100 different E-CLM classifiers were trained on the bioactivity prediction task. The level of confidence in a prediction was defined as the number of models that classified a given input molecule as "highly active".

With increasing confidence levels, the number of molecules predicted as "highly active" decreased (Fig. 3a), a documented effect of ensemble voting[51]. None of the molecules from the focused virtual library was predicted as "highly active" with all 100 votes. Forty-seven de novo designs were predicted as highly active, with 99 votes (Supplementary Figs. 4, 5). Among these top-ranked molecules, 64% featured a new atom scaffold and 62% featured a new graph scaffold with respect to the fine-tuning set (see Supplementary Fig. 6 for exemplary molecule decompositions into graph and atom scaffolds)[52,53]. Higher confidence was reflected in the increased substructure similarity of the predicted actives to the molecules of the fine-tuning set, as captured by the Tanimoto index computed on Morgan fingerprints (Fig. 3b)[54]. In line with the chemical similarity principle[55], this observation suggests that there is a greater chance of identifying active molecules when the number of votes is high. The five most dissimilar molecules among the top-ranked molecules had a similarity to their respective nearest neighbors of the fine-tuning set, ranging from 53 to 58% (Fig. 3c). The closest molecules of the fine-tuning set have a similarity ranging between 77 and 100%, meaning that one molecule of the fine-tuning set was re-created by the CLM, although with a different stereochemistry (Fig. 3d), a structural feature that is not captured by Morgan fingerprints. This result highlights the potential of the approach to explore both closely related molecules to known bioactives, e.g., for structure-activity relationship studies or hit-to-lead expansion, as well as more structurally innovative compounds for "scaffold hopping".

## In vitro bioactivity of commercially available compounds

For a proof of concept, some of the molecules generated by the CLM were tested for PI3Kγ binding in vitro. To optimize the efficiency in terms of both time and resources, we selected the test molecules from the refined virtual chemical libraries that could be purchased from commercial suppliers, as opposed to synthesizing the de novo designs. In total, 16 computer-generated molecules were commercially available. Their predictive confidence ranged from 80/100 votes for compound **1** to 24/100 votes for compound **16** (Fig. 4).

Although none of the ordered molecules was part of the top-ranked set (i.e., receiving 99/100 votes), compound **1**, the molecule with the highest number of votes (80/100), was a hit, with $K_d$ ranging between 0.6 and 0.7 μM ($N = 2$; 670 nM; 620 nM) (Figs. 4, 5 and Supplementary Table 3). None of the lower-ranking compounds inhibited PI3Kγ in the biochemical assay (Fig. 4). The confidence level of our ensemble correctly prioritized compound **1** (active in vitro) over compounds **2** and **4** (inactive in vitro), despite all of them having the same scaffold but with different substituents (Fig. 4). We hypothesize that this might be due to the positive effect of the ELECTRA pretraining, which was aimed at recognizing the effect of small structural changes.

Compound **1** has a new atom scaffold compared to all molecules in the ChEMBL database (version 28) annotated with "pActivity" ≥ 5.0 on PI3Kγ (pActivity: −log(molar $IC_{50}$, $XC_{50}$, $EC_{50}$, $AC_{50}$, $K_i$, $K_d$, or "potency")). It constitutes a "scaffold hop" from known inhibitors. The most similar molecule among these has a Tanimoto similarity of 34% to that of compound **1** (Supplementary Fig. 7). When screening the commercial library with the ensemble of predictive models, compound **1** (80/100 votes) would have appeared in the top 52, showcasing the effect of combining de novo design with scoring. It should be noted that no commercially available molecule received more than 89 votes. The presence of a top-ranking de novo design with 99 out of 100 votes highlights the ability of generative molecular design to explore the chemical space beyond commercially available compounds.

Aiming to benchmark these results, similarity-based virtual screening using the commercial compound library was carried out using Morgan fingerprints (Tanimoto index) and the fine-tuning molecules as queries[56]. Compound **1** ranked in position 25,693, highlighting the ability of our pipeline to uncover a hit that would likely not have been found by chemical structure-based similarity searching. The in vitro validation advocates the E-CLM ensemble prediction approach for ranking the computer-generated molecular designs, aiming to identify bioactive molecules with new scaffolds (core structures).

## Synthesis and bioactivity testing generated molecules

Motivated by the positive results of the scaffold-hopping exercise for hit finding, we synthesized two of the computer-generated top-ranked molecules (**17**, **20**; 99/100 votes) and derivatives thereof (**18**, **19**, **21**, **22**). De novo designs **17** and **20** were selected from all computer-generated molecules receiving 99/100 votes by the E-CLM ensemble model. Aiming to distinguish between these top-scoring molecules we used TIGER software (v19.7, inSili.com LLC) for target prediction and grouped the molecules according to their

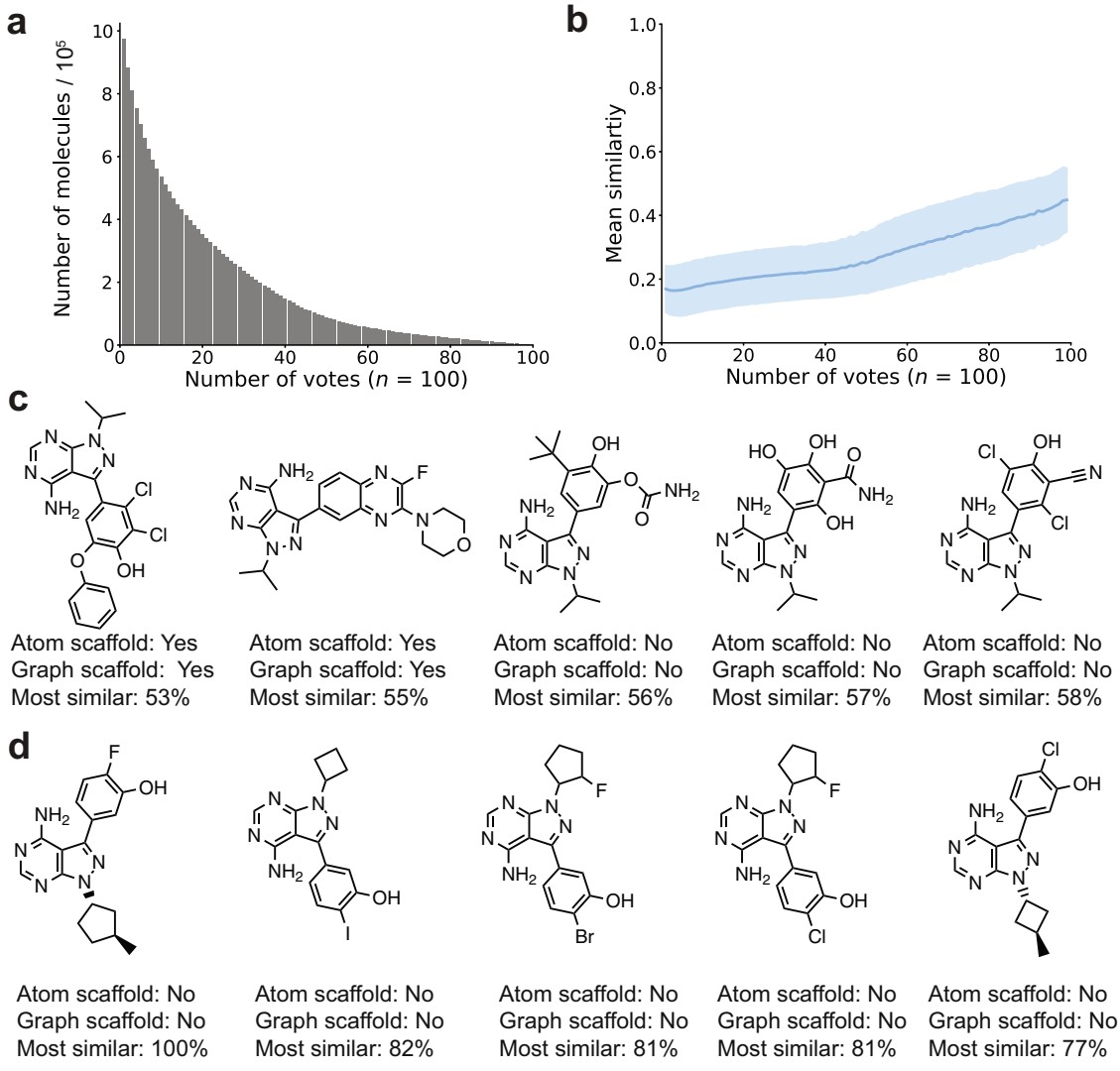

**Fig. 3 | Molecule ranking with a deep ensemble model. a** Number of molecules in the refined virtual chemical library that were predicted as "highly active" as a function of the number of votes (confidence level). **b** Average structural similarity (Tanimoto similarity index computed on Morgan fingerprints) of each de novo design to the fine-tuning set as a function of the number of votes. The solid line represents the mean value, with the shaded area representing the standard deviation. **c** Top-ranked designs (99/100 votes) selected with the most distant nearest neighbor, whose similarity is indicated below the structure ("Most similar") in the fine-tuning set. The atom ("Atom scaffold") and graph ("Graph scaffold") scaffold novelty of the structure with respect to the fine-tuning set is indicated below each structure ("Yes": new, "No": not new). **d** Top-ranked designs (99/100 votes) selected with the closest nearest neighbor in the fine-tuning set.

scaffolds. Scaffold **S1** (Fig. 5) was the most frequently generated core of the de novo designs, for which PI3K binding or inhibition was predicted by TIGER. De novo compounds **17** and **20** received favorable TIGER scores (1.8 and 2.1). They are structurally closely related to the known dual Bruton's tyrosine kinase (Btk) and PI3Kδ inhibitor **23** and PI3Kγ/δ inhibitor **24** (Fig. 5)[57,58]. It is noteworthy that molecules **17**, **20**, **23**, and **24** were not contained in the CLM training or fine-tuning data. The highest similarity of **17** and **20** to compounds from the CLM fine-tuning set were 57 and 63%, respectively. These compounds feature the same pyrazolopyr-imidine kinase hinge binding motif, but the de novo-generated molecules structurally differ in sidechain positions R$_{1-4}$ (Fig. 5). The generative molecule construction method re-created the known generic scaffold **S1** with new sidechain variations. However, ensemble E-CLM scoring alone could not differentiate between compounds **17** and **20**. Both received equally confident votes. We chose this particular example to investigate the applicability of the generative approach to hit expansion rather than hit finding. Such a situation emulates hit-to-lead development in medicinal chemistry.

Molecules **17** and **20** could not be obtained via the preferred retro-synthetic routes suggested by IBM RXN (www.rxn.res.ibm.com)[59]. We, therefore, devised suitable synthesis paths manually. The compounds were afforded in seven and four steps, respectively (Supplementary Information: Chemical Synthesis and Analytics). Purified compounds were tested for direct PI3Kγ binding. The results of this assay revealed potent activity in the nanomolar range ($K_d$ values, expressed as the average of $N = 2$ independent experiments: **17**, 63 nM; **18**, 52 nM; **19**, 160 nM; **20**, 120 nM; **21**, 290 nM; **22**, 13 nM; Supplementary Figs. 8–10). Compounds **17** and **20** were considerably more active than hit compound **1**, which was appropriately reflected in the more confident E-CLM voting (99/100 for **17** and **20**; 80/100 votes for **1**). The most potent compound **22** was devised manually, motivated by the original de novo designs.

Aiming to rationalize the difference in activity between compound **1** and the 4-amino-pyrazolopyrimidine derivatives **17–22**, automated ligand docking was performed using GOLD software[60]. Plausible ligand binding poses in the modeled active site of human PI3Kγ (PDB ID: 3ENE)[57] were obtained for all molecules (Fig. 6).

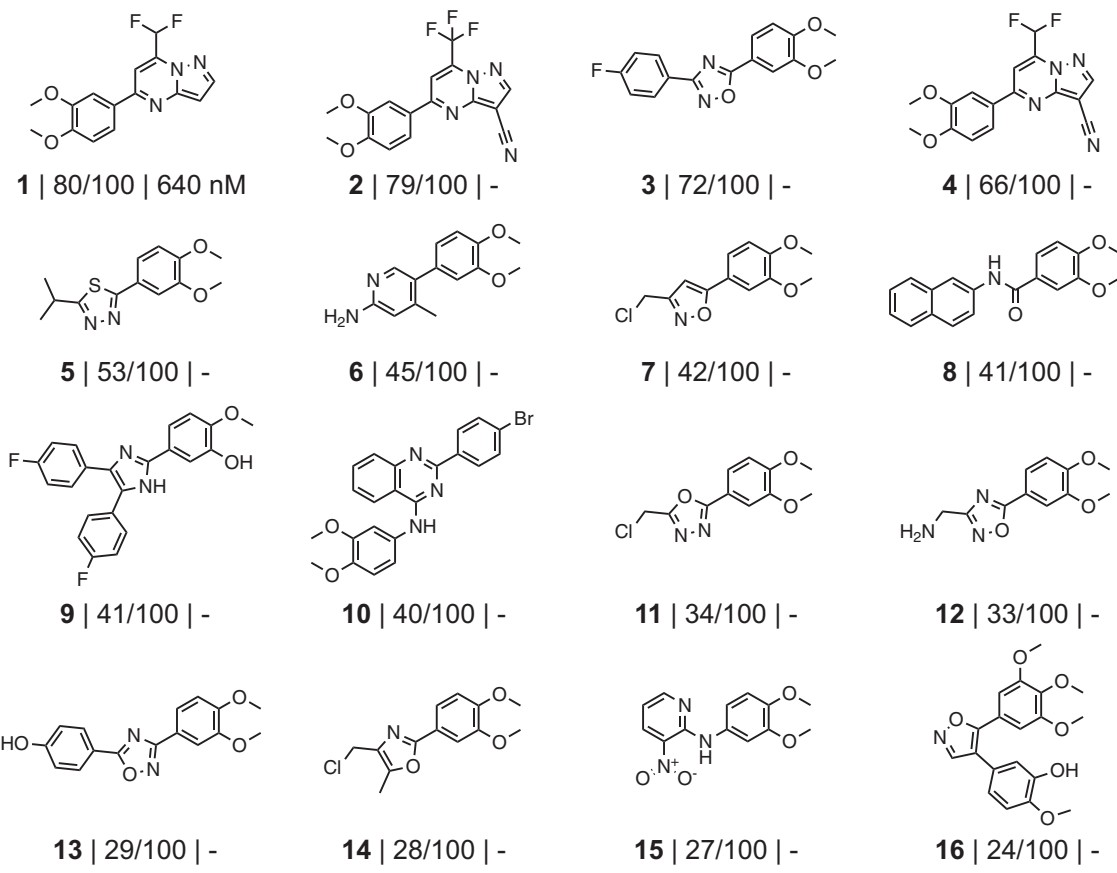

**Fig. 4 | Commercially available compounds tested for PI3Kγ inhibition.** Compounds **1**–**16** are shown, together with the number of votes from the ensemble of the maximum number of 100 possible votes and the experimentally determined binding constant $K_d$. The absence of a value (−) indicates no observed binding of the compound to the target.

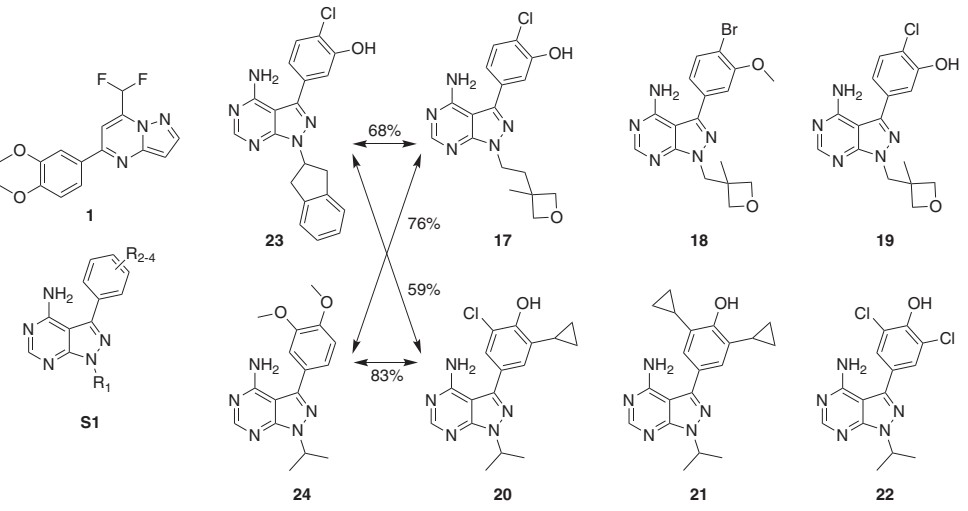

**Fig. 5 | Chemical structures of bioactive ligands.** Computer-generated molecule **1** is a commercially available PI3Kγ ligand identified by the chemical language model. Percent values (Tanimoto coefficient) reflect the similarity of the synthesized computer-generated molecules **17** and **20** to the known PI3K inhibitors **23** and **24**. Compounds **18**, **19** and **21**, **22** are analogs of molecules **17** and **20**, respectively.

Induced-fit docking of molecules **17**–**22** suggested the aromatic and exocyclic nitrogens forming key hydrogen bridges to the kinase hinge residues Glu880 and Val882. Free energies of binding were estimated to range from −29.8 to −34.7 kJ mol⁻¹ for compounds **1** and **17**, respectively (Supplementary Table 4). These values are generally in agreement with the experimental bioactivities, reflecting the relatively weaker kinase binding of compound **1**. The 4-amino-pyrazolopyrimidine derivatives **17**–**22** achieved lower estimated binding energies than compound **1**, highlighting the importance of the hydrogen bridges to the kinase hinge residues Glu880 and Val882. At the same time, the docking results also reveal shortcomings of the quantitative estimation of free energies, which, for this given an

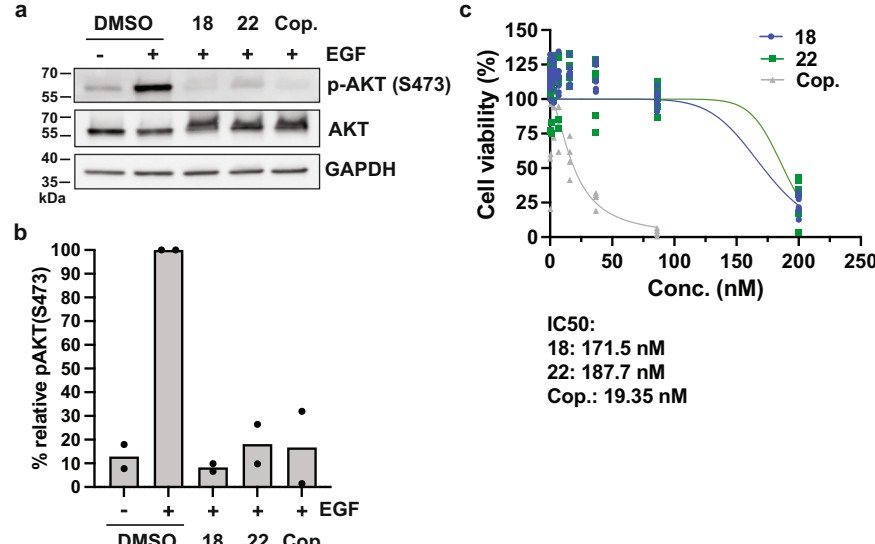

**Fig. 6 | Compounds 18 and 22 repress PI3K-AKT signaling in tumor cells.**
**a** Representative immunoblot analysis p-AKT (S473), AKT, and GAPDH (loading control) using total lysates of human HD-MB03 tumor cells stimulated with 10 ng ml⁻¹ EGF for 15 min in the absence or presence of compounds **18** or **22**, or copanlisib (Cop.). Compounds and copanlisib were used at 100 nM concentration. DMSO was used as solvent control. **b** Quantification of p-AKT levels relative to EGF-stimulated DMSO control. $N = 2$ from two independent experiments. **c** CellTiter-Glo assay to monitor the viability of cells after 72 h of exposure to increasing concentrations of **18**, **22**, or copanlisib in a medium supplemented with FBS. A nonlinear fit of inhibitor versus normalized response is shown. Means (dots) and SD (error bars) of $N = 4$ (copanlisib) or $N = 7$ (compounds **18** or **22**) measurements combined from two independent experiments are shown.

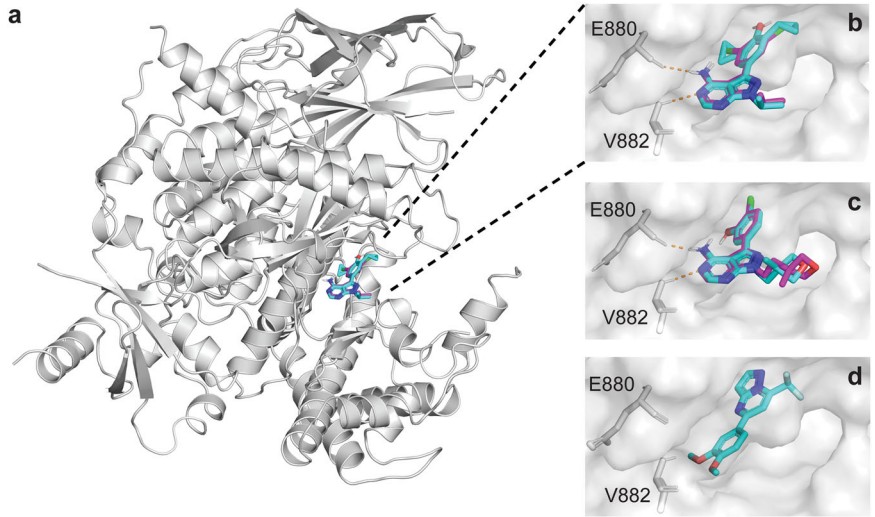

**Fig. 7 | Docking poses of novel PI3Kγ inhibitors. a** Global view of PI3Kγ (PDB ID: 3ENE) with stick models of molecules **21** (blue) and **22** (magenta) docked into the active site of PI3Kγ. **b** Close-up view of the suggested binding poses of molecules **21** (blue) and **22** (magenta). **c** Close-up view of the binding poses of molecules **17** (magenta) and **18** (blue). **d** Close-up view of the computed binding pose of molecule **1** (in blue).

example, failed to correctly rank the 4-amino-pyrazolopyrimidine derivatives **17–22** according to their experimental $K_d$ values (Supplementary Table 4).

Taken together, the results indicate that the molecular design approach presented here could identify both new scaffolds and structural analogs of bioactive compounds for computer-assisted hit finding and expansion. E-CLM ensemble scoring proved applicable to virtual ligand screening but could not differentiate between structurally closely related potent ligands. Using an external scoring function (TIGER software) for target prediction proved useful in this study, complementing the twin-CLM approach.

To confirm the biological activities of the most potent compounds **18** ($K_d = 52$ nM) and **22** ($K_d = 13$ nM) in cells, we tested compound effects on epidermal growth factor receptor (EGFR)-induced

activation of AKT/protein kinase B (PKB), a well-established effector downstream of EGFR-PI3K signaling in cancer[61]. In response to growth factor receptor and PI3K activation, AKT is phosphorylated on Ser473 by the Rictor-mTOR complex[62]. Thus, the phosphorylation of AKT on Ser473 can be used as a surrogate read-out for PI3K activity in cells. Treatment of the serum-starved human brain tumor cell line HD-MB03 with epidermal growth factor (EGF), the physiological ligand of the EGFR, leads to the phosphorylation of AKTS473 (Fig. 7a and Supplementary Fig. 11). Pre-treatment of the cells with **18** or **22** at 100 nM concentration prevents AKTS473 phosphorylation (Fig. 7a) and causes an ~70–90% reduction in AKT phosphorylation levels (Fig. 7b). The reduction in EGF-induced AKT phosphorylation by compounds **18** and **22** is comparable to the inhibition caused by an equimolar concentration of the pan-PI3Kα/β/γ/δ inhibitor copanlisib (BAY 80-6946)

with reported sub-nanomolar inhibitory activity against PI3Kα and PI3Kδ (PI3Kγ $IC_{50}$ = 6.5 nM)[63]. In our medulloblastoma cell-based assay, compounds **18**, **22**, and copanlisib had $IC_{50}$ values of ~172, 188, and 19 nM, respectively. To exclude the possibility that reduced AKT phosphorylation caused by compound **18** and **22** treatments is the result of reduced viability or cell death, we monitored the viability of compound-treated cells using the CellTiter-Glo assay. We found that neither **18** nor **22** caused a significant reduction in viability at 100 nM concentration (Fig. 7c), thus excluding an indirect effect on AKT function due to toxicity. The predicted best-fit $IC_{50}$ values in the CellTiter-Glo assay for compounds **18**, **22**, and copanlisib are ~172, 188, and 19 nM, respectively. In conclusion, both compounds **18** and **22** repress the activation of the PI3K effector kinase AKT under physiologically relevant conditions in human tumor cells at a low nanomolar concentration.

De novo drug design aims to generate new molecular scaffolds with desired properties and, at the same time, suggest more subtle structural modifications for hit-to-lead optimization. The results of this study positively advocate the CLM approach for both design tasks. Methodological improvements in CLM training advanced the sampling of target-focused virtual molecule libraries. The hybrid CLM classifier included both structural and bioactivity information of the fine-tuning molecules for the design of a virtual chemical library, thereby complementing the available methodological repertoire for virtual screening. It remains to be determined in more detail to what extent the CLM pretraining method affects model performance in the downstream task, i.e., molecular generation or ordinal classification. Importantly, CLM training was performed without data augmentation to study the positive effect of nucleus sampling on the generation of a SMILES string. Future improvement might be possible by combining nucleus sampling with data augmentation for CLM transfer learning[33,64–66]. Given the setup of the present study, it was not possible to determine whether our hypothesis regarding the beneficial effect of model pretraining on patented chemical structures holds true. The long time required for hit-to-lead expansion and preclinical and clinical drug development until a marketed drug is obtained will likely preclude such an analysis. Irrespectively, three de novo designs were successfully tested for bioactivity. Hit compounds **1** (80/100 votes), **17**, and **20** (99/100 votes) were correctly predicted to be active by CLM ensemble scoring. The difference in votes is reflected in the high nanomolar and low nanomolar dissociation constants, respectively. Obtaining rapid experimental validation of a set of readily available de novo-designed molecules prior to embarking on de novo synthesis might help assess the value of computationally generated activity-focused chemical libraries. Future prospective studies will also have to assess the general applicability of this approach to other drug targets from different target families, and improve the prediction accuracy of the CLM classifier for structurally closely related molecules. Furthermore, although in this study we only considered in vitro bioactivity (direct binding and cellular activity), the workflow could be further extended to consider the multi-dimensional nature of drug discovery projects (e.g., membrane permeability, aqueous solubility, and off-target activity) in a data-driven manner. Drawing any conclusions as to its superiority to other methods or broad applicability would be premature. Additional prospective studies will be necessary to help answer this question. Whether a particular de novo design method may be considered "better" than another one critically depends on the specific task, rendering any general method evaluation challenging. We consider practical application the best judge and a way forward.

This study highlights the versatility of generative deep learning for hit and lead finding in drug discovery. The computational pipeline can be used to both create new molecules and screen libraries of existing compounds. We envision future projects in which de novo design methods are first validated for physically available molecules from a compound repository or commercial suppliers before investing in more expensive and time-consuming syntheses of computer-generated molecules. This strategy could potentially help accelerate the design-make-test-analyze cycle of drug discovery[67].

## Methods

### Target selection
The protein target PI3Kγ[26] was selected on the basis of the data available in the DTC[35] database. We selected one of the targets with the highest number of annotated data points. Molecules with activity entries satisfying each of the following conditions were kept (standard relation: "=", standard unit: "nM", substrate value: "10", substrate unit: "μM", test inhibitor type: "competitive inhibitor", compound concentration value: "0.001–50", test assay format: "biochemical", test assay type: "functional", test assay subtype: "enzyme activity"). This filtering step resulted in a dataset containing 198 molecules. Duplicate entries of small molecules with orders of magnitude difference in their reported activity were deleted (Supplementary Fig. 1).

### Training data
The training molecules were represented as canonical SMILES strings using the RDKit package (v. 2019.03.2, https://www.rdkit.org). SMILES strings with a length of up to 90 characters were retained and standardized in Python (v. 3.6.5) by removing salts and duplicates. The CLM was pretrained on the pharmaceutical subset of the US patent database[34,68]. After the processing, 839,674 unique molecules encoded as canonical SMILES strings constituted the pretraining data. PI3Kγ inhibitors with reported bioactivity ≤100 nM in the DTC database were used for the CLM transfer learning ("fine-tuning set"). This criterion resulted in a fine-tuning set containing 43 molecules.

### CLM pretraining and fine-tuning for the generation of SMILES strings
The CLM model was implemented in Python (v. 3.6.5) using Keras (v. 2.2.0, https://keras.io/) with the TensorFlow GPU backend (v. 1.9.0, https://www.tensorflow.org). The model was implemented as a recurrent neural network with LSTM cells. The neural network was composed of four layers with a total of 5,820,515 parameters (layer 1: BatchNormalization, layer 2: 1024 LSTM cells, layer 3: 256 LSTM cells, and layer 4: BatchNormalization) and trained with SMILES data encoded as one-hot vectors. The CLM was trained using the Adam optimizer (learning rate = $10^{-3}$) and the categorical cross-entropy loss function. The training was performed over 40 epochs, where one epoch was defined as one pass over all the training data. Transfer learning was performed by keeping the parameters of the first network layer constant and training the second layer with a learning rate of $10^{-4}$.

### ELECTRA pretraining
The E-CLM model was implemented in Python (v. 3.6.5) using Keras (v. 2.2.0, https://keras.io/) with the TensorFlow GPU backend (v. 1.9.0, https://www.tensorflow.org). The ELECTRA model was implemented with the same architecture as that of the generative CLM, i.e., as a recurrent neural network with LSTM cells. Model training was performed with the Adam optimizer (learning rate = $10^{-3}$, 50 epochs) and the binary cross-entropy loss function.

### Ordinal classifier training
The hybrid CLM network contained the weights of the pretrained E-CLM plus an additional feedforward layer with three sigmoidal neurons. The model was trained to solve an ordinal classification task, where each of the three output neurons corresponded to one class. $k$-Means clustering ($k = 5$, Scikit-learn; https://scikit-learn.org/stable/) was performed to group the fine-tuning molecules according to their similarity based on Morgan fingerprints. Four groups were used for cross-validation and one for classifier testing. The output threshold values, the number of transfer learning epochs, and the oversampling

values of the less represented classes were defined by cross-validation. The best settings were selected based on the performance on the test set, which was used once (oversampling: +40 molecules for the two less represented classes, sigmoid threshold: 0.4, number of transfer learning epochs: 200). Each of the 100 CLM models of the final ensemble was trained with the best settings on all available data. The neural network architecture was composed of six layers with a total of 5,646,982 parameters (layer 1: BatchNormalization, layer 2: 1024 LSTM cells, layer 3: 256 LSTM cells, layer 4: BatchNormalization, layer 5: Dropout, and layer 6: Dense, with three units, each with a sigmoid activation function) and was trained with SMILES encoded as one-hot vectors. The models were trained with the Adam optimizer and the binary cross-entropy loss function (learning rate $= 10^{-4}$, 200 epochs).

## Temperature sampling
SMILES characters were sampled using the softmax function parameterized by the sampling temperature. The probability of the $i$-th character being sampled from the CLM predictions was computed as (Eq. 1)

$$q_i = \exp(z_i/T)\Big/\sum_j \exp(z_j/T), \tag{1}$$

where $z_i$ is the CLM prediction for character $i$, $T$ is the temperature, and $q_i$ is the sampling probability of character $i$.

## Nucleus sampling
SMILES characters were sampled with a temperature value equal to 1 (Eq. 2), considering only characters whose cumulative probability was greater than the nucleus parameter ("top vocabulary"):

$$\sum_{x \in V^{(p)}} P(x \mid x_{1:i-1}) > p, \tag{2}$$

where $V^{(p)}$ is the top vocabulary, $x$ is an element of the vocabulary, and $p$ is the nucleus parameter.

## Commercial compound library screening
Molecules were ranked based on their similarity (Tanimoto index, Morgan fingerprints) to the molecules used to fine-tune the generative CLM. As a fusion rule, the reciprocal sum of ranks was calculated to obtain a score value, $S$, for each molecule (Eq. 3)[56]:

$$S = \sum_{i=1}^{N} \frac{1}{\text{rank}(x_i)}, \tag{3}$$

where $i$ runs over all $N$ fine-tuning molecules, and $\text{rank}(x_i)$ is the rank obtained from the similarity between the considered design and the $i$-th molecule of the fine-tuning set (the higher the similarity, the higher the rank). Greater $S$ values correspond to better rank positions in the fused list of molecules.

## Automated ligand docking
The crystal structure of human PI3Kγ (PDB ID: 3ENE) was retrieved from the Protein Data Bank (https://www.rcsb.org/) and prepared with MOE v.2019.0102 (Chemical Computing Group, Montreal, Canada) with the following settings: QuickPrep module: "Preserve Sequence and Neutralize"; "Use Protonate 3D for Protonation' = True; 'Allow ASN/GLN/HIS "Flips" in Protonate 3D' = True; 'Delete Water Molecules Farther than 4.5 Å from Ligand or Receptor" = True, Tether Receptor: Strength = 10, Buffer = 0.25; Fix: "Atoms Farther than 8 Å from Ligands", hydrogens close to ligands not fixed; Refine: "to RMS Gradient of 0.1 kcal/mol/Å"; "Retain QuickPrep Minimization Restraints" = True. Ligand molecules were docked with GOLD v.5.2.2 within MOE v.2019.0102 (Chemical Computing Group, Montreal, Canada): Efficiency = default, Score efficiency = 100; Early

Termination = [number:3, RMS = 1.5], GOLD scoring, Induced Fit, 80 poses per compound. Poses were refined with MOE GBVI/WSA dG (40 refinement poses)[69], and the top-scoring pose of each compound was selected for further analysis. The applied GoldScore function GBVI/WSA dG estimated free binding energies based on four components: (i) protein–ligand hydrogen bond energy, (ii) protein–ligand van der Waals energy, (iii) ligand internal van der Waals energy, and (iv) ligand torsional strain energy. Redocking of the crystalized small molecule ligand (PDB ID: 3ENE, 1-methyl-3-naphthalen-2-yl-1H-pyrazolo[3,4-d] pyrimidin-4-amine) yielded a root mean square deviation of 0.448 Å and an estimated binding energy of −33.93 kJ mol$^{-1}$.

## Biochemical kinase binding assay
PI3Kγ binding assays were performed by Eurofins Discovery (https://www.eurofinsdiscoveryservices.com) on a fee-for-service basis. KINOMEscan™ was used to determine the dissociation constant $K_d$. The assay was based on the ability of a test compound to compete with an immobilized active site-directed ligand. Competition of the test compound with the immobilized ligand was measured via quantitative PCR (qPCR) of the DNA tag of DNA-tagged kinase[70]. An 11-point three-fold serial dilution of each test compound was prepared in 100% DMSO at 100× final test concentration and subsequently diluted to 1× in the assay (final DMSO concentration = 1%). Dissociation constants were estimated with a standard dose-response curve using the Hill equation (Eq. 4)[71]:

$$\text{Response} = \text{Background} + \frac{\text{Signal} - \text{Background}}{1 + \left(K_d^{\text{Hill slope}}/\text{Dose}^{\text{HillSlope}}\right)}, \tag{4}$$

where the Hill slope was set to −1. Curves were fitted using a nonlinear least square fit with the Levenberg–Marquardt algorithm[72,73].

## Cells and cell culture for in-cell activity analysis
HD-MBO3 Group 3 medulloblastoma cells were cultured in RPMI medium supplemented with 10% fetal bovine serum (FBS, both from Sigma-Aldrich, St. Louis, USA), 1% Penicillin-Streptomycin, and 1% GlutaMAX (both from Gibco/Thermo Fisher Scientific, Waltham, USA). The cells were maintained at 37 °C in a humidified atmosphere containing 5% CO$_2$. Chemicals: Copanlisib (BAY 80-6946, Selleckchem, Munich, Germany) and other compounds were dissolved in 100% dimethyl sulfoxide (DMSO) at a stock concentration of 10 to 100 mM and stored at −20 °C. DMSO was used as solvent control in all assays.

## Immunoblotting
The cells were seeded in six-well plates and starved in a serum-free medium for 48 h before treatment. Cells were pretreated with either DMSO (solvent control), 100 nM copanlisib (positive control) or compounds for 3 h, and then treated for 15 min with 10 ng ml$^{-1}$ recombinant human EGF protein (PeproTech, London, UK, Cat.100-15). Cells were then lysed using RIPA buffer (30 mM HEPES, pH 7.4, 150 mM NaCl, 1% Nonidet P-40, 0.5% sodium deoxycholate, 0.1% sodium dodecyl sulfate, 5 mM EDTA) supplemented with protease (Complete Mini) and phosphatase inhibitors (PhosSTOP, both from Roche, Basel, Switzerland) and cleared by centrifugation for 5 min. Protein concentration was measured using the Pierce BCA Protein Assay Kit according to the manufacturer's (Thermo Fisher Scientific, Waltham, USA) instructions. Protein separation was performed on Mini-Protean TGX (4–20%) SDS-PAGE gel and transferred to PVDF membranes (both from Bio-Rad, Hercules, USA). After 1 h of blocking with 5% non-fat dry milk, membranes were probed with primary anti-AKT (#9272), anti-phospho-AKT (p-AktSer473) (#4060), and anti-GAPDH antibodies (#2118, all from Cell Signaling Technology, Danvers, USA,). HRP-linked secondary antibodies were used to detect the primary antibodies. Chemiluminescence detection was carried out using ChemiDoc Touch Gel and Western Blot imaging system (Bio-Rad,

Hercules, USA). Integrated densities of protein bands from two independent IB experiments were determined using Fiji software[74]. Background signal was subtracted from all measurements. p-AKT signal was normalized to total AKT.

### Cell viability/proliferation (Cell TiterGlo) assay
Cell viability was determined using CellTiter-Glo® 2.0 Cell Viability Assay (G9242, Promega, Madison, USA). About 500 cells per 20 µl were seeded in a flat bottom 384-well plate (781098, Greiner bio-one, Kremsmünster, Austria). Increasing concentrations of compounds were deposited on cells using an HP Digital Drug Dispenser (Hewlett-Packard, Palo Alto, USA) with DMSO total volume normalization. After 72 h, the CellTiter-Glo reagent was added (volume/volume) following the manufacturer's instructions. Luminescence representing the number of viable cells was quantified with a Cytation 3 imaging reader (BioTek, Winooski, USA). Measurements from a total of $N = 4$ (copanlisib) or 7 (compounds **18** or **22**) technical replicas were combined from two independent experiments. Prism 9 software (GraphPad Software, San Diego, CA, USA) was used to calculate best-fit values of inhibitor vs. normalized response and to predict $IC_{50}$.

### Chemical synthesis and analytics
For a full description of the chemical synthesis and compound analytics, see the Supplementary Material.

### TIGER target prediction
TIGER software (version 19.7, inSili.com GmbH, Zurich, Switzerland) was used as described in ref. 75. Molecules were represented in terms of CopyCATS (version 3.2) topological pharmacophore descriptor vectors for input to TIGER[76]. Any predictions of PI3K subtypes were considered correct predictions.

### Reporting summary
Further information on research design is available in the Nature Portfolio Reporting Summary linked to this article.

## Data availability
The training data used in this study are available from Zenodo at https://zenodo.org/record/7370858 (https://doi.org/10.5281/zenodo.7370858).

## Code availability
The computational framework presented in this study and the pretrained neural network weights are available from Zenodo at https://zenodo.org/record/7370858 (https://doi.org/10.5281/zenodo.7370858).

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

## Acknowledgements

This work was financially supported by the Swiss National Science Foundation (grants no. 205321_182176 to G.S. and no. CRSII5_202245/1 RaSMID to G.S. and M.B.) and by the RETHINK initiative at ETH Zurich. The authors thank Alexander Button, Damian Gautschi, and Jonas Bossart for helpful discussions. Daniel Lowe and Lukas Friedrich are thanked for their technical assistance with the US patent database.

## Author contributions

M.M., F.G., and G.S. conceived the study. M.M. implemented the software. C.B. curated the data. K.A. and I.P.A. performed the docking study. I.P.A. and L.C. synthesized the compounds. S.Y. and M.B. designed the cell-based assays. S.Y. performed the cell-based assays. G.S. supervised the study. All authors analyzed the data and results and contributed to the writing of the manuscript.

## Competing interests

G.S. declares a potential financial conflict of interest as a consultant to the pharmaceutical industry and co-founder of inSili.com GmbH, Zurich, Switzerland. The authors declare no other competing interests.
