## [Peer Review File · Nature Communications]

Reviewer comments, first round -

Reviewer #1 (Remarks to the Author):

This manuscript describes an important methodology for de novo generation of molecules with the desired bioactivity using Chemical Language Model (CLM). All methodological and computational aspects are described clearly, and the authors should be commended for sharing their data and approaches via GitHub. However, this reviewer has serious reservations about the practical value of the proposed approach for drug discovery based on the way the results of the study are presented as some of those appear confusing.

The most important issue is the novelty of the tested compound 1 and the ability of the described methodology to produce novel bioactive compounds with novel scaffolds assessed both independently and in comparison, with alternative, simpler methodologies. In general, the authors do a good job of justifying the need to place restrictions on the diversity of de novo generated molecules as compared to the training set. However, such restriction obviously reduces the novelty. From the presented results, it is not clear what one should expect in terms of external prediction accuracy. Given that the authors choose commercially available compounds to validate their designs, it would be important to benchmark their approach versus virtual screening of the same database using the respective QSAR model that they used to bias library generation. It is certainly nice that one newly identified compound showed activity in respective assay, but how can the authors explain the lack of activity of other selected and tested molecules? To add to this uncertainty that unfortunately is not addressed by the authors, hit compound 1 is presented as a novel molecule with an atomic scaffold not found in ChEMBL; yet supplemental figure S4 shows the respective chemical structure annotated as having pChEMBL activity just like the other three molecules in Figure S4. So, was this compound actually found in ChEMBL?

In addition to this critical major point, there are several other issues that need to be addressed as discussed below.

- The sources of data are poorly annotated. The authors should provide direct identifiers for compounds and assays in respective databases, i.e., ChEMBL and DTC. For instance, please provide compound ChEMBL IDs in Figure S4; provide a list of IDs for known PI3K inhibitors used for activity classification model.
- Depending on the argument, the authors switch between FCD and Tanimoto to estimate compound similarity: what is the reason for such switching?
- In the beginning, the authors put forward the hypothesis about the use of US Patent Database versus ChEMBL to generate more drug like molecules: was it confirmed? What criteria were used to reach the definitive conclusion?
- Abstract: the main hit compound is described as sub-micromolar; but the text refers to it as nanomolar. Given that the measured activity is nearly 1 μ M, calling it submicromolar sounds appropriate.
- P 2, top paragraph: Unclear sentence: "inhibitors of vascular endothelial growth factor receptor 2 kinase and the unfolded protein response pathway". Are there any inhibitors of the unfolded protein response pathway?
- P3: "However, with this "temperature sampling" approach, SMILES characters are unlikely to be sampled..": What is the meaning of this statement? I think the authors are trying to imply that resulting SMILES will not have the desired activity, but the sentence as written appears rather senseless.
- Supplementary figure S1 is supposed to show how filtering of DTC resulted in "a dataset containing 198 molecules"; however, this figure shows 6 molecules with the largest distribution of PIC50 values. Authors should address this inconsistency.

Reviewer: Alex Tropsha

Reviewer #2 (Remarks to the Author):

This manuscript by Moret et al. reports a molecule generative model for designing PI3Kgamma inhibitors. The topic is of interest and there have been high expectations that generative modeling

may improve our abilities to design new medicines. Here, a language model is trained based a corpus of existing molecules in the format of SMILES. The model encodes the molecular representation and can be decoded back to sample new molecules, which was used to generate focused chemical library biased toward known PI3Kgamma inhibitors. The idea is not new but the methodological part has novelty, where the ELECTRA approach was used for improving the pretraining to better capture minor structural changes, and a downstream task of bioactivity prediction was incorporated for selecting candidate molecules. The manuscript has been well organized and clearly written. I have the following suggestions for the authors' consideration to improve the manuscript.

Major:

1. It is a pity that the authors didn't select the highest ranked molecules to rigorously evaluate the AI model through synthesis and biological assays. From the analysis of modeling results, the false positive rate of the model is as low as 10%, but unfortunately the experimental verification results are not satisfactory: only one hit was successfully identified among 16 purchased chemicals, which is not much different from that of a conventional virtual screening campaign. As put forward by Patrick Walters (Nat Biotechnol. 2020; 38:143-145), I strongly recommend that the authors should synthesize and evaluate the best predicted molecules, and the results are crucial to support the significance of the computational model.
2. This manuscript combines both computational and experimental studies, but the experimental findings are less significant. It is true that a novel scaffold PI3Kgamma inhibitor was identified, but I think identifying weakly potent ATP-site directed protein kinase inhibitors through virtual screening is no longer considered a significant advance in this maturing field because of the availability of many known potent inhibitors acting by this mechanism. More data about the developability such as kinase selectivity or PK profiles of this molecule is needed.
3. There have been many pretrained models for molecule generation based on different molecular representations, the authors should compare their method with other benchmark models to show its advantages in generating new/active molecules.

Reviewer #3 (Remarks to the Author):

The manuscript titled "Leveraging molecular structure and bio activity with chemical language models for drug design" describes application of methods first applied to natural language processing and adapted for chemical design. They employ a commonly used approach of pretraining a model to learn a predefined chemical space of interest and the syntax of SMILES. Then the model is refined for chemical structure generation on a subset of data that is more specific for a certain task, in this case PI3ky binding assay.

It is refreshing that the authors are taking a stance of testing generated compounds and ,while they constrained their testing space to purchasable compounds, this is a movement in the right direction to drive adoption of molecular generation methods. It is also very appreciated that the code is provided. The manuscript is suggested to be published with only a few minor points addressed.

Comments:

- The candor of the authors in the conclusion in regard to shortcomings and next steps is appreciated, one more comment that should be made somewhere in the manuscript (and I hope the authors are moving towards in further research) is that drug discovery is not a single objective optimization but is rather a very complex multicomponent optimization.
- In the SI add a figure analogous to Figure 4 showing the top 20-30 compounds regardless of their ability to be purchased. In the current form the reader only has ~5 examples from figure 3 which all have the exact same core heterocycle (NC1=C(C=NN2)C2=NC=N1). Are the top 20-30 just more permutations or are there some other core heterocycles.
- Can the authors add a few examples of the outcomes to Murcko decompositions using RDKit to the SI with a pointer in the main text. For non-programmers and bench chemists, they might have seen many different versions using different toolkits. Eg for a molecule in Figure 3C NC1=C(C(C2=C(Cl)C(Cl)=C(O)C(OC3=CC=CC=C3)=C2)=NN4C(C)C)C4=NC=N1 RDKit gives the

core as: c1ccc(Oc2cccc(-c3n[nH]c4ncncc34)c2)cc and Openeye gives:
c1ccccc1.c1ccccc1.c1c2cn[nH]c2ncn1 as the result. It is important so that the reader understand the specificity of what a "core" is defined to be by the toolkit used to decompose the molecules. This definition is very important for the reader to understand since a lot of the analysis and discussion is based on it. Obviously using the openeye toolkit (from this example) would result in more overlap. No need for comparisons of toolkits since many are closed source, just some examples of the ones you did use.

- What is the rationale for the cutoff for filtering the inhibitors from the DTC?

Response to the reviewers

We thank all three expert reviewers for their constructive criticism. Peer review at its best!

Reviewer #1 (Remarks to the Author):

This manuscript describes an important methodology for de novo generation of molecules with the desired bioactivity using Chemical Language Model (CLM). All methodological and computational aspects are described clearly, and the authors should be commended for sharing their data and approaches via GitHub. However, this reviewer has serious reservations about the practical value of the proposed approach for drug discovery based on the way the results of the study are presented as some of those appear confusing.

The most important issue is the novelty of the tested compound 1 and the ability of the described methodology to produce novel bioactive compounds with novel scaffolds assessed both independently and in comparison, with alternative, simpler methodologies. In general, the authors do a good job of justifying the need to place restrictions on the diversity of de novo generated molecules as compared to the training set. However, such restriction obviously reduces the novelty.

>> For clarification, please note that we did not impose any restrictions on the diversity of the de novo designs. All molecules were generated in a rule-free manner by the CLM and automatically prioritized based on the numbers of votes. The top-scoring compound has a similarity towards known bioactive molecules below 32%, as computed with the Tanimoto coefficient on Morgan fingerprints (Supplementary Fig. S7). This molecule features a novel scaffold. Moreover, given the modularity of the pipeline and the high number of molecules that one could in theory generate, the user has the option to choose structurally more or less diverse molecules (e.g., for hit finding or hit-to-lead expansion) among the designs. Since the primary aim of our study was to assess the usefulness of the E-CLM molecule scoring system and our new twin-CLM method, we did not apply such considerations on diversity to rule out the introduction of confounding factors in the evaluation. This aspect of chemical diversity has been addressed in more detail in the revised version of the manuscript.

From the presented results, it is not clear what one should expect in terms of external prediction accuracy. Given that the authors choose commercially available compounds to validate their designs, it would be important to benchmark their approach versus virtual screening of the same database using the respective QSAR model that they used to bias library generation.

>> Commercially available compounds were selected to validate the de novo computational approach with fewer resources. Thus it is not directly comparable with conventional virtual screening, which does not generate new molecular entities. Nonetheless, to address this reviewer's comment, we screened the commercial library both with the predictive ensemble CLM, as well as with a well-established and simpler method (Tanimoto similarity based on Morgan fingerprints). Only with the new CLM-based QSAR model, compound 1 would have been found among the top-ranking 52 molecules. Of note, the top-scoring molecule on this ranked list

received only 89 out of 100 votes, highlighting the benefit of generating a de novo library focused on the target of interest (n.b. some of the de novo molecules received 99/100 votes). When screening the commercial library with the Tanimoto similarity based on Morgan fingerprints, compound 1 was ranked in position 25,693, showing that it would most likely not have been selected for in vitro analysis with such a conventional approach. We have revised the manuscript accordingly to incorporate these results.

It is certainly nice that one newly identified compound showed activity in respective assay, but how can the authors explain the lack of activity of other selected and tested molecules?

>> One of the goals of this experiment was to define an “activity threshold” for the votes of the E-CLM ensemble. Ideally, one would select only the molecules in the highest-ranked list (i.e., the molecules with 99 positive votes out of 100), as one would expect the highest success rate in this list. The top-ranked compounds that were available in the commercial library screened received “only” 80 positive votes, while the lowest-ranked compound received 24 votes. In essence, the positive result we found with 80 votes helps set the bar for the number of votes that are sufficient for finding positive results. With these results, it now seems reasonable to invest the time and resources to synthesize the highest-ranked compounds (i.e., with 99 positive votes out of 100). Indeed, we synthesized and tested two such de novo generated compounds and four derivatives. These all had nanomolar inhibitory activity, fully corroborating the scoring approach. Please see the accordingly revised and updated manuscript.

To add to this uncertainty that unfortunately is not addressed by the authors, hit compound 1 is presented as a novel molecule with an atomic scaffold not found in ChEMBL; yet supplemental figure S4 shows the respective chemical structure annotated as having pChEMBL activity just like the other three molecules in Figure S4. So, was this compound actually found in ChEMBL?

>> The compound was not found in ChEMBL. It is a novel chemical structure. We used the term “pChEMBL” instead of “pKi OR pKd” for the sake of simplicity in comparison with molecules reported in ChEMBL. We agree that this might be confusing. Thus, we have revised Supplementary Fig. S4 (now labeled as Supplementary Fig. S7) to remove this ambiguity.

In addition to this critical major point, there are several other issues that need to be addressed as discussed below.

- The sources of data are poorly annotated. The authors should provide direct identifiers for compounds and assays in respective databases, i.e., ChEMBL and DTC. For instance, please provide compound ChEMBL IDs in Figure S4; provide a list of IDs for known PKI3 inhibitors used for activity classification model.

>> We thank the reviewer for this comment. For the bioactivity data used to train the classification model, we have added a .csv file to the project's GitHub repository (https://github.com/michael1788/hybridCLMs/blob/main/data/bioactivity_data_P48736.csv). It contains the identifiers, along with all other assay annotations. We added the ChEMBL IDs to Figure S4 (now Figure S7).

- Depending on the argument, the authors switch between FCD and Tanimoto to estimate compound similarity: what is the reason for such switching?

>> 5,000 to 10,000 molecules are required to compute the FCD as reported by the authors of the original publication. Thus we could not use it to assess the compound similarity during the fine-tuning phase as the fine-tuning set contains only 43 molecules (“low-data” scenario).

- In the beginning, the authors put forward the hypothesis about the use of US Patent Database versus ChEMBL to generate more drug like molecules: was it confirmed? What criteria were used to reach the definitive conclusion?

>> Several studies [e.g., Segler, M. H., Kogej, T., Tyrchan, C., & Waller, M. P. (2018). Generating focused molecule libraries for drug discovery with recurrent neural networks. *ACS central science*, 4(1), 120-131., Moret, M., Friedrich, L., Grisoni, F., Merk, D., & Schneider, G. (2020). Generative molecular design in low data regimes. *Nature Machine Intelligence*, 2(3), 171-180. Grisoni, F., Huisman, B. J., Button, A. L., Moret, M., Atz, K., Merk, D., & Schneider, G. (2021). Combining generative artificial intelligence and on-chip synthesis for de novo drug design. *Science Advances*, 7(24), eabg3338.] have shown that statistically significant inductive bias can be introduced by properly choosing the molecules to include in the pretraining set. These properties will be ‘inherited’ also in the corresponding designs, e.g., fraction of sp³-hybridized carbons, predicted synthesizability, and substructure elements. Of course, as highlighted in the conclusions section, it is not possible to verify if the corresponding designs are indeed more drug-like, due to the complexity of capturing drug-likeness and to the long time/cost associated to determine drug-likeness experimentally. Following the reviewer’s comment, we rephrased our initial statement to improve clarity.

- Abstract: the main hit compound is described as sub-micromolar; but the text refers to it as nanomolar. Given that the measured activity is nearly 1 μM, calling it submicromolar sounds appropriate.

>> Agreed. We updated from nanomolar to sub-micromolar in the main text. Please note that the new results indeed contain low-nanomolar hits.

- P 2, top paragraph: Unclear sentence: “inhibitors of vascular endothelial growth factor receptor 2 kinase and the unfolded protein response pathway”. Are there any inhibitors of the unfolded protein response pathway?

>> We rephrased the sentence in the revised version of the manuscript to address this point.

- P3: “However, with this “temperature sampling” approach, SMILES characters are unlikely to be sampled..”: What is the meaning of this statement? I think the authors are trying to imply that resulting SMILES will not have the desired activity, but the sentence as written appears rather senseless.

>> Thank you for pointing this out. We rephrased the sentence accordingly.

- Supplementary figure S1 is supposed to show how filtering of DTC resulted in “a dataset containing 198 molecules”; however, this figure shows 6 molecules with the largest distribution of PIC50 values. Authors should address this inconsistency.

>> Thank you for pointing this out. We addressed this inconsistency in the revised manuscript.

Reviewer #2 (Remarks to the Author):

This manuscript by Moret et al. reports a molecule generative model for designing PI3Kgamma inhibitors. The topic is of interest and there have been high expectations that generative modeling may improve our abilities to design new medicines. Here, a language model is trained based a corpus of existing molecules in the format of SMILES. The model encodes the molecular representation and can be decoded back to sample new molecules, which was used to generate focused chemical library biased toward known PI3Kgamma inhibitors. The idea is not new but the methodological part has novelty, where the ELECTRA approach was used for improving the pretraining to better capture minor structural changes, and a downstream task of bioactivity prediction was incorporated for selecting candidate molecules. The manuscript has been well organized and clearly written. I have the following suggestions for the authors' consideration to improve the manuscript.

Major:

1. It is a pity that the authors didn't select the highest ranked molecules to rigorously evaluate the AI model through synthesis and biological assays. From the analysis of modeling results, the false positive rate of the model is as low as 10%, but unfortunately the experimental verification results are not satisfactory: only one hit was successfully identified among 16 purchased chemicals, which is not much different from that of a conventional virtual screening campaign. As put forward by Patrick Walters (Nat Biotechnol. 2020; 38:143–145), I strongly recommend that the authors should synthesize and evaluate the best predicted molecules, and the results are crucial to support the significance of the computational model.

>> Thanks for this comment. We certainly agree. For the revision, we selected two of the top-scoring computer-generated molecules and several derivatives thereof, which were synthesized and tested in vitro. All of these compounds (99/100 votes) had nanomolar activity. These new results demonstrate that the de novo design and scoring methodology is suitable for both hit finding (new scaffolds) and hit-to-lead expansion (high-scoring designs). These results have been incorporated in the revised version of the manuscript.

2. This manuscript combines both computational and experimental studies, but the experimental findings are less significant. It is true that a novel scaffold PI3Kgamma inhibitor was identified, but I think identifying weakly potent ATP-site directed protein kinase inhibitors through virtual screening is no longer considered a significant advance in this maturing field because of the availability of many known potent inhibitors acting by this mechanism. More data about the developability such as kinase selectivity or PK profiles of this molecule is needed.

>> The scope of this study is to propose a computational pipeline for automatic design and ranking of new molecules possessing desired properties. We did not primarily aim to identify novel potential drug candidates for PI3K γ . This is why we (i) set out to have minimal interference on the molecule selection, and base it solely on the model output, and (ii) assessed molecules based on the properties modeled by the computational pipeline (i.e., predicted bioactivity on the chosen target rather than selectivity or PK profiles). Purchasing of compounds did not allow us to explore the regions of the chemical space with the highest predictions (99/100 votes), because none of these were commercially available, but nonetheless allowed us to find a bioactive molecule. This preliminary result provided insights on the suitability of the method to identify bioactive molecules

and gave confidence to devote more resources and synthesize de novo designs with the top scores. Accordingly, we synthesized and tested two of the top-scoring “de novo generated” molecules plus several derivatives thereof. Potent nanomolar PI3K γ inhibition by these compounds was observed. The scaffold of these compounds is similar to known PI3K γ inhibitors (leading to high predictive confidence) but at the same time the design algorithm introduced new side chains and functionalities. These additional experimental results validate the approach for both hit identification and hit expansion.

3. There have been many pretrained models for molecule generation based on different molecular representations, the authors should compare their method with other benchmark models to show its advantages in generating new/active molecules.

>> We based our study on one of the current state-of-the-art models for molecule generation according to diverse benchmarks (e.g., Brown, Nathan, et al. "GuacaMol: benchmarking models for de novo molecular design." *Journal of chemical information and modeling* 59.3 (2019): 1096-1108. and Flam-Shepherd, D., Zhu, K., & Aspuru-Guzik, A. (2021). Keeping it Simple: Language Models can learn Complex Molecular Distributions. arXiv preprint arXiv:2112.03041.).

Reviewer #3 (Remarks to the Author):

The manuscript titled “Leveraging molecular structure and bio activity with chemical language models for drug design” describes application of methods first applied to natural language processing and adapted for chemical design. They employ a commonly used approach of pretraining a model to learn a predefined chemical space of interest and the syntax of SMILES. Then the model is refined for chemical structure generation on a subset of data that is more specific for a certain task, in this case PI3ky binding assay.

It is refreshing that the authors are taking a stance of testing generated compounds and ,while they constrained their testing space to purchasable compounds, this is a movement in the right direction to drive adoption of molecular generation methods. It is also very appreciated that the code is provided. The manuscript is suggested to be published with only a few minor points addressed.

Comments:

- The candor of the authors in the conclusion in regard to shortcomings and next steps is appreciated, one more comment that should be made somewhere in the manuscript (and I hope the authors are moving towards in further research) is that drug discovery is not a single objective optimization but is rather a very complex multicomponent optimization.

>> We certainly agree with the reviewer's comment and have rewritten the conclusion in the revised manuscript accordingly to address this important consideration.

- In the SI add a figure analogous to Figure 4 showing the top 20-30 compounds regardless of their ability to be purchased. In the current form the reader only has ~5 examples from figure 3 which all have the exact same core heterocycle (NC1=C(C=NN2)C2=NC=N1). Are the top 20-30 just more permutations or are there some other core heterocycles.

>> We added two figures to the SI (Supplementary Fig. S4 and S5) with the highest-ranked compounds, i.e., the 47 molecules with 99 of 100 possible positive votes.

- Can the authors add a few examples of the outcomes to Murcko decompositions using RDKit to the SI with a pointer in the main text. For non-programmers and bench chemists, they might have seen many different versions using different toolkits. Eg for a molecule in Figure 3C NC1=C(C(C2=C(CI)C(CI)=C(O)C(OC3=CC=CC=C3)=C2)=NN4C(C)C)C4=NC=N1 RDKit gives the core as: c1ccc(Oc2cccc(-c3n[nH]c4ncncc34)c2)cc and Openeye gives: c1ccccc1.c1ccccc1.c1c2cn[nH]c2ncn1 as the result. It is important so that the reader understand the specificity of what a “core” is defined to be by the toolkit used to decompose the molecules. This definition is very important for the reader to understand since a lot of the analysis and discussion is based on it. Obviously using the openeye toolkit (from this example) would result in more overlap. No need for comparisons of toolkits since many are closed source, just some examples of the ones you did use.

>> We have added three examples of atom and graph scaffold decompositions from a molecule with RDKit to the SI (Supplementary Fig. S6).

- What is the rationale for the cutoff for filtering the inhibitors from the DTC?

>> The chosen cutoffs correspond to commonly used guidelines when considering molecules as bioactive hits. For bioactivity predictions, cutoff filtering at a pIC₅₀ of 4.0 allowed us to include inactive molecules, which augmented the data for model training. The cutoff value of 6.5 for fine-tuning and molecule selection was used to consider only “highly active” molecules, which are usually the target of de novo design.

Reviewer comments, second round -

Reviewer #1 (Remarks to the Author):

This reviewer is satisfied with the answers; it is especially gratifying to see that while working on the revisions, the authors discovered more active compounds.

Reviewer #2 (Remarks to the Author):

In this revised manuscript, the authors added the synthesis and evaluation of two new compounds designed by the CML model, which showed molecular-level of activity in the nanomolar range. The author's efforts in this regard should be acknowledged, but the new data does not better support the key argument of the paper in any way. What we expect on the CML models is to obtain new chemotypes that break away from the traditional medicinal modification ideas, or, in the authors' word, to automatic design and ranking of new molecules possessing desired properties. Obviously, the newly synthesized two compounds have existing kinase inhibitor scaffolds, and such compounds can hardly be called "new molecules". We can easily obtain such molecules using traditional rule-based generative methods or conventional medicinal transformation and substitution of functional groups.

In general, my opinion on this work still holds. This manuscript combines both computational and experimental studies, the computational part lacks solid data support, and the experimental findings are trivial. In addition, the authors replied that they have added the performance comparison with other deep generation methods, but I did not find the corresponding content in the revised manuscript.

Reviewer #3 (Remarks to the Author):

The authors have addressed the questions sufficiently and this report is suggested to be accepted

Response to the reviewers

Reviewer #2 (Remarks to the Author):

In this revised manuscript, the authors added the synthesis and evaluation of two new compounds designed by the CML model, which showed molecular-level of activity in the nanomolar range. The author's efforts in this regard should be acknowledged, but the new data does not better support the key argument of the paper in any way.

> The choice of which decisive data to add was actually suggested by this reviewer in the previous reviewing round: “It is a pity that the authors didn't select the highest ranked molecules to rigorously evaluate the AI model through synthesis and biological assays. From the analysis of modeling results, the false positive rate of the model is as low as 10%, but unfortunately the experimental verification results are not satisfactory: only one hit was successfully identified among 16 purchased chemicals, which is not much different from that of a conventional virtual screening campaign. As put forward by Patrick Walters (Nat Biotechnol. 2020; 38:143–145), I strongly recommend that the authors should synthesize and evaluate the best predicted molecules, and the results are crucial to support the significance of the computational model.” We followed this reviewer's suggestion accordingly. In response to this comment, we selected, synthesized and biologically tested two of the top-ranked molecules (plus four of their derivatives as a cautionary measure to eliminate the possibility of a chance result). The computer-generated molecules are in perfect agreement with the hit rate of our model, as confirmed by the experimental results. In fact, both compounds (and the four derivatives) were potentially active, showing activity values consistent with the computational predictions. Therefore, we consider these new data to answer the point raised by the reviewer.

What we expect on the CML models is to obtain new chemotypes that break away from the traditional medicinal modification ideas, or, in the authors' word, to automatic design and ranking of new molecules possessing desired properties.

> “De novo” means “from scratch” (“autonomously”). De novo design algorithms (CLMs or other approaches) aim to capture medicinal chemistry knowledge. A useful tool should be able to autonomously generate synthesizable molecules with desired properties. Probing this capacity of the CLM approach was a goal of our study. The results positively advocate the applicability of the new machine learning pipeline to hit finding (new scaffold, compound 1) and scaffold decoration (compounds 17, 19). While we fully agree with this reviewer that “novel” chemotypes can be desirable, depending on the state of a drug discovery project, it is well known that deep learning models have limited capability for out-of-distribution generalization. Systematically finding new chemotypes with a model based on SMILES strings (which capture explicit information on atom types and their connectivity) would mean to generalize out of the training domain. At no point do we claim to achieve this goal with the method presented. Importantly, the virtual screening example demonstrated the ability of the CLM approach to actually generate new bioactive chemotypes, as exemplified with compound 1. The additional compounds we synthesized and tested are top-scoring computer-generated molecules (as requested by the reviewer), which, by definition — and due to the inner functioning of the neural networks

employed — are the most similar molecules to the training data. We achieved a 100% hit rate with these de novo designs.

Obviously, the newly synthesized two compounds have existing kinase inhibitor scaffolds, and such compounds can hardly be called "new molecules". We can easily obtain such molecules using traditional rule-based generative methods or conventional medicinal transformation and substitution of functional groups.

> The molecules are factually *new*, although not necessarily *innovative*, because they were generated from scratch by the software without human interference. This result demonstrates that the machine learning pipeline had captured relevant structure-activity relationships from the training data in a rule-free manner. Moreover, as clearly stated in the manuscript, the chosen application was designed for proof-of-concept and did not aim to identify novel patentable drugs in the first place. We wrote (p.10): "We chose this particular [second] example to investigate the applicability of the generative approach to hit expansion rather than hit finding." Although the most confident predictions were made for the molecules close to the training set, end-to-end machine learning pipelines (like the one presented here) are more flexible than conventional medchem pipelines or rule-based approaches as they allow for (i) tuning the uncertainty/chemical diversity ratio, (ii) exploring the chemical space without being bound to molecular construction rules, and (iii) applications to new targets for which no structure-activity relationship is known and limited data is available.

Addressing the reviewer's second statement, we are glad to learn that this reviewer considers the computer-generated molecules indistinguishable from human designs. This statement further corroborates the practical applicability of the proposed machine learning pipeline. Importantly, by considering only "easy" modifications of a known hit (N.B., "easy" is this reviewer's subjective opinion rather than an objective assessment of compound structure or synthesizability), the number of combinatorial possibilities by far exceeds the number of molecules one could actually synthesize and test in a drug discovery project. Therefore, one should not only ask if a particular molecular structure (e.g., scaffold, functionalization) could have been suggested by a medicinal chemist, but how many compounds a given model needs to actually find a suitable lead. Exhaustively addressing this question exceeds the scope of the present study as it will require numerous additional studies by the whole research community.

In general, my opinion on this work still holds. This manuscript combines both computational and experimental studies, the computational part lacks solid data support, and the experimental findings are trivial.

> We respectfully disagree with the reviewer. In particular:

1. Data support to the computational part: We introduce a new experimental pipeline, with both a new pre-training scheme (ELECTRA) and molecule sampling (nucleus) approach. None of these approaches have ever been combined with CLMs. We compared nucleus sampling for 10 repeats, 1000 molecules each, for 20 epochs, and 18 threshold values (i.e., more than 3M SMILES strings) to the common temperature sampling method. Several metrics were used to this end, as recommended by established guidelines (Brown et al. "GuacaMol: benchmarking models for de novo molecular design." *Journal of Chemical Information and Modeling* 59, 1096-1108 (2019)). For model pre-training, we thoroughly compared the ELECTRA method to the

widely employed autoregressive approaches on more than 800,000 molecules, and measured the effect of two different training strategies. The computed hit rates were corroborated by the wet-lab experiments. This, in our opinion, constitutes solid evidence to support the computational part of our study.

2. Experimental findings: The results fully corroborate the validity of the computational pipeline in terms of its capability to generate bioactive molecules and identify promising molecules. Importantly, this is one of the select few published studies in which de novo designs proposed by a computational model were experimentally validated. The most potent compounds led to pronounced inhibition of PI3K-dependent Akt phosphorylation in a cell model of medulloblastoma, comparable to the marketed drug copanlisib, which we consider not trivial to obtain.

In addition, the authors replied that they have added the performance comparison with other deep generation methods, but I did not find the corresponding content in the revised manuscript. > Possibly, the reviewer misread or misinterpreted our response. The reviewer might have overlooked that we performed a rigorous comparison of different training and sampling methods as described in the main text. (Note our original response to this reviewer's comment: "We based our study on one of the current state-of-the-art models for molecule generation according to diverse benchmarks (e.g., Brown, et al. GuacaMol: benchmarking models for de novo molecular design." *Journal of chemical information and modeling* 59.3 (2019): 1096-1108. and Flam-Shepherd, D., Zhu, K., & Aspuru-Guzik, A. (2021). Keeping it Simple: Language Models can learn Complex Molecular Distributions. *arXiv preprint arXiv:2112.03041*.)"